# Coupled Simulation of Hydrate-Bearing and Overburden Sedimentary Layers to Study Hydrate Dissociation and Methane Leakage

**Yan Xie [1,2], Jingchun Feng [1,2,\*], Liwei Sun [1,2], Junwen Wang [1,2], Weiqiang Hu [1,2], Bo Peng [3], Yujun Wang [3] and Yi Wang [4,5]**

[1] Research Centre of Ecology & Environment for Coastal Area and Deep Sea, Guangdong University of Technology, Guangzhou 510006, China; yanxie@gdut.edu.cn (Y.X.); liweisun@gdut.edu.cn (L.S.); 2112124014@mail2.gdut.edu.cn (J.W.); 2112124041@mail2.gdut.edu.cn (W.H.)
[2] Southern Marine Science and Engineering, Guangdong Laboratory (Guangzhou), Guangzhou 511458, China
[3] Guangdong Eco-Engineering Polytechnic, Guangzhou 510520, China; pengbo@gig.ac.cn (B.P.); wangyujun0023@163.com (Y.W.)
[4] Key Laboratory of Gas Hydrate, Guangzhou Institute of Energy Conversion, Chinese Academy of Sciences, Guangzhou 510640, China; wangyi@ms.giec.ac.cn
[5] Guangzhou Center for Gas Hydrate Research, Chinese Academy of Sciences, Guangzhou 510640, China
\* Correspondence: fengjc@gdut.edu.cn; Fax: +86-020-3932-2141

**Abstract:** Methane leakage during natural gas hydrate (NGH) exploitation is one of the important challenges restricting its safe development, which necessitates further investigation. However, only a few experimental studies have been conducted to characterize the relationship between methane ($CH_4$) leakage and NGH exploitation. The $CH_4$ leakage mechanism and controlling factors in the hydrate dissociation process are still unclear. A coupled simulator has been developed to study the $CH_4$ hydrate exploitation and the possible leakage of $CH_4$. The new system overcomes the difficulty of constructing hydrate-free overlying strata and seawater in previous studies and can simulate the in situ natural environment containing hydrate reservoirs, overlying strata and overlying seawater as well. In addition, the simulator integrates the spatial distribution of temperature, pressure and electric resistance in hydrate reservoir systems, and allows for the visual monitoring of the overlying strata and the sampling of overburden gas and liquids. The effectiveness of the coupled simulations was verified through experimental testing. The coupled simulations allowed for the characterization of the $CH_4$ leakage mechanism and can be used to develop safe strategies for NGH exploitation.

**Keywords:** natural gas hydrate; methane leakage; $CH_4$ hydrate dissociation; simulator; dynamic separation

## 1. Introduction

Natural gas hydrate (NGH), also known as combustible ice, is a non-stoichiometric crystalline compound, in which water molecules form cages and trap gas molecules [1]. The formation of NGH requires a suitable low temperature, high pressure and sufficient gas source. NGHs in ocean environments account for 97% of the global hydrate resources [2]. $CH_4$ is the main gas component in NGH, and about 160–170 volumes of $CH_4$ gas can be reserved in each volume of NGH. It is widely accepted that NGH is the largest source of hydrocarbon on earth, even though multiple estimations of its reserves differ by several orders of magnitude [3]. Due to the characteristics of large reserves, high energy density and almost no pollution after combustion, NGH has attracted the attention of a large number of scholars.

Since the mid-1990s, several field tests of gas production from NGH have been carried out [4–6], and two field tests have been conducted in the South China Sea [7]. However, the large-scale commercial development of NGH still faces a series of bottlenecks. In addition

to the sand production and productivity problems, geological and environmental disasters such as reservoir collapse and methane leakage may occur during gas production from NGH [8], which is also one of the problems most worried about by scientists.

As a huge $CH_4$ reservoir in the earth, NGH plays an important role in the methane balance and carbon cycle of the earth [9,10]. $CH_4$ leakage from only 0.01% of global NGH dissociation could lead to serious disasters, including ocean acidification, biological extinction and climate warming [11,12]. Many geological evidences have shown that NGH is closely related to climate change [13]. Generally, the natural dissociation of NGH mainly results from the rise of marine temperature or geological movement [14]. However, drilling, depressurization and other activities during hydrate exploitation can directly trigger the hydrate dissociation, which may result into blowout, and above mentioned collapse and $CH_4$ leakage [15]. Compared to the accompanied $CH_4$ leakage problem in shale gas extraction [16], the exploitation of NGH involves not only the leakage of free gas, but also the transient instability and local overpressure due to hydrate dissociation, which is a higher-risk gas production behavior.

In order to realize the NGH development, while limited by the high risk and cost of the field test, an increasing number of the hydrate exploitation researches have been conducted in laboratory with the methods mainly including depressurization [17–20], heat stimulation [21–24], inhibitor injection [25–27], $CO_2$ replacement [28–31] and their combination. In the meantime, with the gradual deepening of research, the laboratorial simulation device for hydrate exploitation has been developed from small to large, and the detection means from simplicity to diversification and accuracy [32,33]. However, most of the current devices can only simulate the hydrate-bearing reservoir within closed system, which is mainly used to study the hydrate accumulation law and efficient gas production method, while rarely considers the influence of the overlying strata and overlying seawater on hydrate dissociation. Moreover, the environmental effects from overlying strata collapse and methane leakage caused by hydrate dissociation are almost ignored. The numerical simulation results showed that influenced by pressure reduction amplitude, geomechanics and overburden pressure, the horizontal well underwent shear failure, and the stratum around the vertical subsided can induce the formation of gas leakage channel [34]. Moridis [35] numerically analyzed the depressurization and hydrate dissociation processes, indicating that although a higher depressurization amplitude increased the driving force for hydrate dissociation, it can also result in the destruction of stratum strength. Therefore, the $CH_4$ leakage law and prevention strategy need to be further clarified to guarantee the security of NGH exploitation.

This work describes a new device (dissociation and leakage coupled simulator, DLCS) focused on $CH_4$ hydrate dissociation and its associated methane leakage behaviors. By using this device, the in situ natural environment containing hydrate-bearing sediment, overlying strata and seawater can be well simulated. The separation and natural contact between the hydrate-bearing reservoir and the overlying strata, respectively, in the hydrate formation and dissociation process could come true by utilizing a mobile separation system. In addition to the spatial distributions of temperature, pressure and electric resistance assembled in hydrate reservoir, the real-time monitoring for the evolution of the overlying strata and the rising process of $CH_4$ gas bubbles could be achieved. Through the combination of these methods, whether methane leakage occurs can be accurately judged. In order to validate the good function of each system of the device, an effective experiment was carried out. As far as we know, this is also the first set of experimental devices used for methane leakage research during hydrate dissociation process.

## 2. Apparatus and Methods

### 2.1. Apparatus Description

The experimental system mainly consists of three parts: reaction, injection and gas production system. The reaction system is the core part of the entire experimental system, which was used to construct the in situ environment containing hydrate sediment, overlying

strata and overlying seawater in the DLCS. The maximum simulated ocean depth of the DLCS can reach 2000 m. According to the operating and experimental requirements of the DLCS, corresponding collaborative systems were equipped.

### 2.1.1. Reaction System

The reaction system mainly includes the high-pressure reactor (DLCS) and a supporting temperature control system. We selected 316 L forgings with corrosion resistance as the steel for the construction of the reactor. Comparing 316 L with conventional cold rolling, the material is more compact and has higher compressive strength. The reactor is mainly composed of an upper cap, a bottom plate, an intermediate reactor body and movable flashboard components. The appearance of the DLCS is shown in Figure 1. The outer chamber of the flashboard is linked with the reactor body by welding, and the upper cap and bottom plate are linked with the body through screws. The effective volume of the whole reactor is about 30.2 L and the maximum pressure resistance is 25 MPa. Due to the great dimension of real NGH deposits, hydrate simulator is gradually developed to large scale through increasing the effective volume. According to reports, the largest reactor has been developed to 1710 L. With the enlarged scale, relatively reduced boundary effect, and more complex hydrate formation/dissociation state, the experimental condition could be regarded as closer to the in situ hydrate-bearing sediment environment [10,33,36]. For the newly built simulator in this study, considering the increasing difficulty of design and experimental operation with the enlargement of the size, the effective volume of 30.2 L was adopted.

The longitudinal section of the reactor is displayed in Figure 2. Based on the movable flashboard, the whole reactor could be divided into an upper overlying chamber (OC) and a lower sediment chamber (SC). The effective volumes of the OC and SC are 19.9 L and 10.3 L, respectively, in which the volume of the upper cap is 8.5 L. The maximum sediment filling height of the SC is 15 cm, the height of the intermediate reactor body contained in the OC is 20 cm, and the filling height of the overlying strata in the OC in this study was 10 cm. The structure of the flashboard system is the core innovation of this device and also the most difficult part to design. According to previous device reports on hydrate exploitation, there is almost no simulated hydrate-free overlying strata above hydrate-bearing sediments, and the key difficulty is the realization of the dynamic separation and natural contact between hydrate sediments and overlying strata. The flashboard in the device is shown in Figure 3, which is composed of a stainless-steel plate with a thickness of 0.5 cm and a movably connected steel rod. The other side of the rod is located in the atmospheric environment outside the reactor system and is connected with a control handle to realize the insertion and extraction of the flashboard. When the board is pulled out from the reactor, it is located in the flashboard chamber, and the upper and lower parts of the reactor are linked. The middle reactor body located in the OC and SC of are internally connected with a spiral sleeve to realize the seal of the two sides of the inserted flashboard through extrusion. In the first few constructions of the separation system, we tried some more complex structures, but the results were not ideal. Finally, we simplified the structure by adopting some fine design and appropriate sealing and realized the dynamic separation of the DLCS. However, due to the thin thickness of the board, the pressure difference between its two sides cannot be too large, otherwise deformation may occur. Therefore, we designed a pressure tracking system to maintain the constant differential pressure and solved this problem. In the experiment, the pressure of the OC of the reactor was generally slightly higher than that in the SC. More details about the pressure equilibrium are described below.

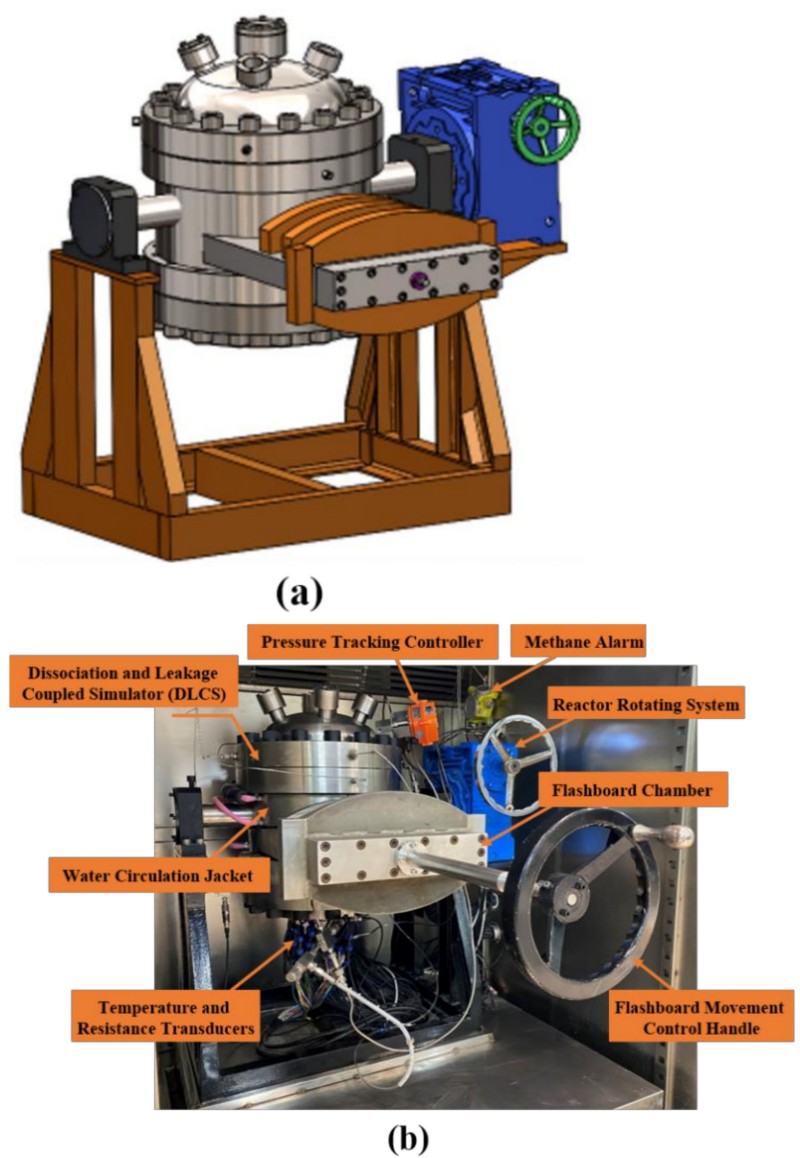

**Figure 1.** Schematic diagram of the DLCS. (**a**) 3D model drawing, (**b**) physical drawing.

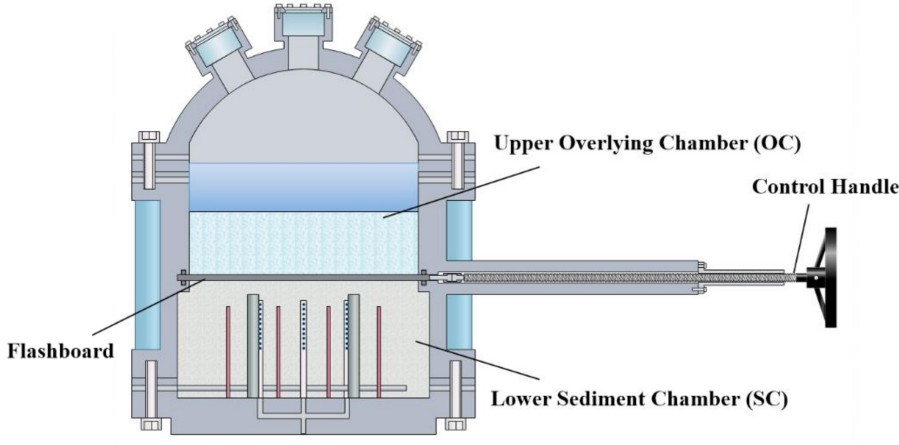

**Figure 2.** Schematic diagram of the DLCS sectional view.

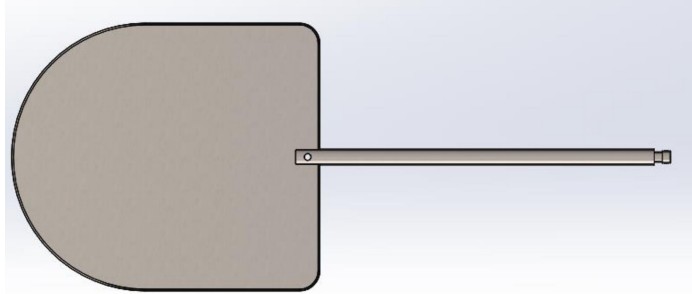

**Figure 3.** The movable flashboard for dynamic separation and natural contact of hydrate-bearing sediment and its overlying strata.

The structure of the bottom plate is shown in Figure 4. The temperature transducers, electric resistance transducers, production wells and heating rods are all embedded into the DLCS from the bottom of the plate. The thermometers are Pt100 with the range of −20~200 °C and measurement accuracy of ±0.1 °C. The electric resistance is measured by Agilent 34970A. The measurement range is 0~1000 KΩ and the accuracy is ±0.1 KΩ; 4 × 4 pairs of temperature and electric resistance transducers are inserted into the bottom, corresponding to the sixteen holes in Figure 4. There is a set of temperature and electric resistance transducers in each of the hole, and the interval between adjacent temperature or electric resistance sensors is 6 cm. For each set of temperature or electric resistance transducers, there are three measuring points in the longitudinal direction, and the interval between adjacent points is 4 cm. Therefore, there are 16 temperature and electric resistance measurement points on each layer, and 48 temperature and electric resistance measurement points distribute in the simulated hydrate sediment space. Through the temperature and electric resistance responses in the hydrate formation/dissociation processes, these measurement points can be used to judge the hydrate position, hydrate state and the spatial dissociation difference in the hydrate exploitation process. Five vertical wells and a horizontal well were arranged in the plate. The production grooves on all the production wells are perforated and wrapped with sand control mesh. Through the selection of the wells, the influence of single well, multi well and well layout on hydrate dissociation can be investigated. In addition, four heating rods with heating power up to 1 kW are installed at the bottom of the plate, which can be used for temperature heating to simulate varied stratum temperature differences or hydrate dissociation by utilizing electric heating method.

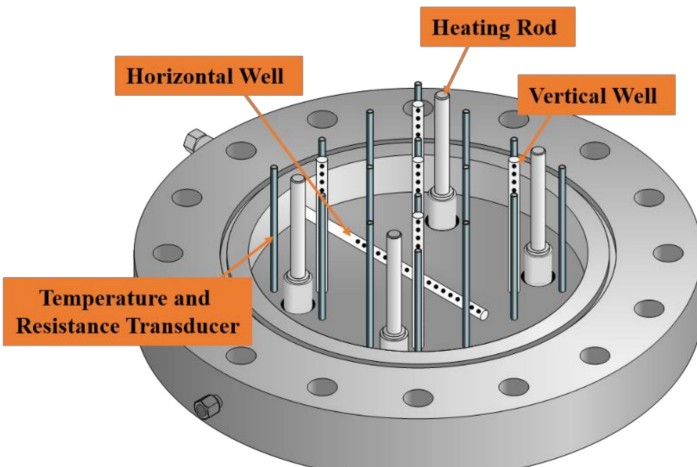

**Figure 4.** The bottom plate of the DLCS inserted with sixteen temperature and electric resistance transducers, five vertical wells, a horizontal well and four heating rods.

The structure of the reactor cap installed with five windows is shown in Figure 5. The central window is mainly used to observe the state changes of overlying strata and water in the OC, and the process is recorded by a charge coupled device (CCD). The four lateral windows are mainly used to provide light source and can also be used for visual observation. In particular, in order to realize the overall record of the simulated overlying layers, the height of the central window structure needs to be designed shorter to expand the observation field. Therefore, sapphire is used as the transparent window, which has strong pressure resistance and good optical performance. In addition, seawater and gas sampling ports are set at the side of the cap to measure the $CH_4$ concentration change in liquid and gas phase.

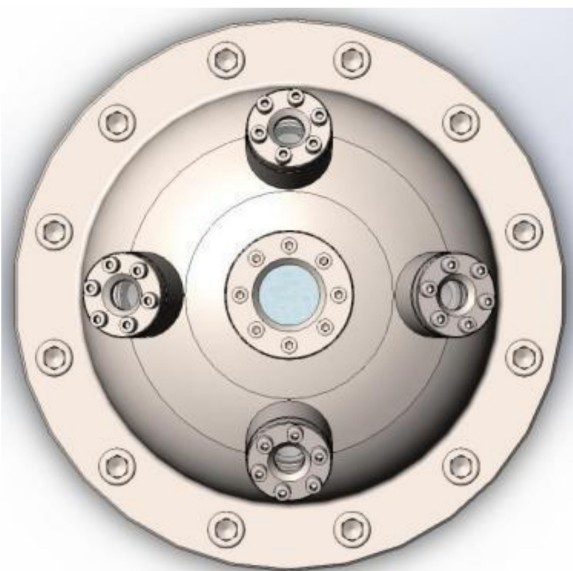

**Figure 5.** Schematic diagram of reactor cap.

To ensure the stability and safety of the reaction system, some detailed controls and connections were designed. As mentioned above, to solve the differential pressure problem of the flashboard, a pressure tracking system was adopted, as shown in Figure 1b. The pressure tracking controller is linked to the reactor cap and controls the automatic gas exhaust of the OC by monitoring the pressure difference between the OC and SC. The pressure difference between the two parts is generally not more than 0.3 MPa in our design program. A gas acquisition port is connected in parallel with the pressure tracking controller to realize the automatic update of the collected gas phase and reduce the measured error caused by inhomogeneous gas phase distribution. In addition, the problem that the hydrate may block the pressure sensor was considered. Four pressure sensors (0~25 MPa, $\pm$0.1 MPa, Trafag) are installed in the SC and mainly at the outlet of the production wells. Due to many of the perforations on production wells, the probability of their complete blockage by hydrate can be decreased. The feedback between the pressure tracking system and the multiple pressure sensors reduces the occurrence of excessive exhaust or non-exhaust of the OC. In view of the overpressure problems that may be caused by hydrate dissociation, a safety valve is connected in the upper and lower cavities of the reactor, and a methane detector is installed in the air bath. In addition, a set of rotation system is configured to realize the 360° rotation of the reactor in the vertical space, which is convenient for the installation, disassembly and maintenance of the reactor system.

The experimental system temperature is controlled by an air bath and a water bath circulation system, as shown in Figure 6. The temperature control ranges of air bath and water bath are both $-10\sim100$ °C, and the accuracy are $\pm0.2$ °C. Through the dual temperature control, the temperature in the reactor can be highly stable and reach the target temperature faster. The air bath mainly provides a constant ambient temperature for the

reaction system, and the water bath circulation system can enhance the heat supply. In addition, the influence of ambient temperature heat transfer on hydrate dissociation in the hydrate exploitation process can be investigated through the change of single experimental conditions and coordinated temperature control.

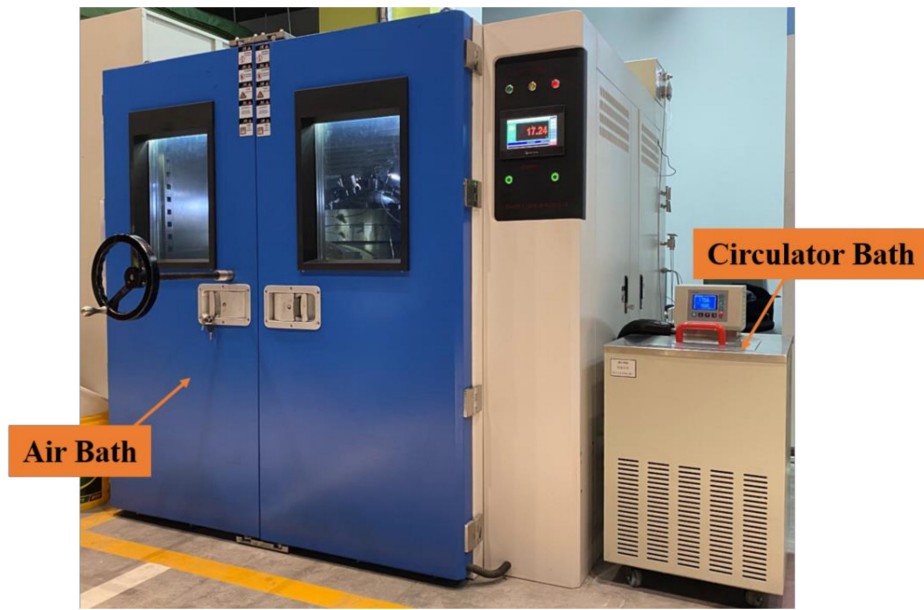

**Figure 6.** Air bath and water bath for DLCS temperature control.

### 2.1.2. Gas-Liquid Supply System

The gas supply system is used for the providing of experimental gas. It is mainly composed of gas booster pump, gas storage tanks and connecting pipeline. Its maximum pressurization can reach 30 MPa, which meets the pressure requirement for the experiment. The liquid injection system mainly depends on a liquid plunger pump, with a maximum pressurization of 25 MPa. The constant speed injection of liquid can be realized by using the plunger pump, and the flow rate range is 0.1–250 mL/min.

### 2.1.3. Gas-Liquid Collection System

The gas-liquid collection system is used for the quantification of gas and liquid in the experimental process, as shown in Figure 7. It is mainly composed of programmed pressure controller, gas-liquid separator, electronic balance, gas dryer and gas flowmeter. The pressure controller plays the role of back pressure with the maximum working pressure up to 30 MPa. By using the pressure controller, the direct depressurization method for hydrate dissociation can be carried out by setting a constant pressure. It can also realize the pressure gradient drop or constant speed drop in the DLCS for hydrate dissociation through its internal program control, in which the minimum depressurization amplitude is 0.01 MPa/min. The gas and water from the program pressure controller enter the separator, and the water yield is recorded in real time through the electronic balance. The gas leaving the separation tank enters the gas dryer, in which the desiccant is anhydrous silica gel. The dried gas enters the gas flowmeter to measure the instantaneous and cumulative gas flows.

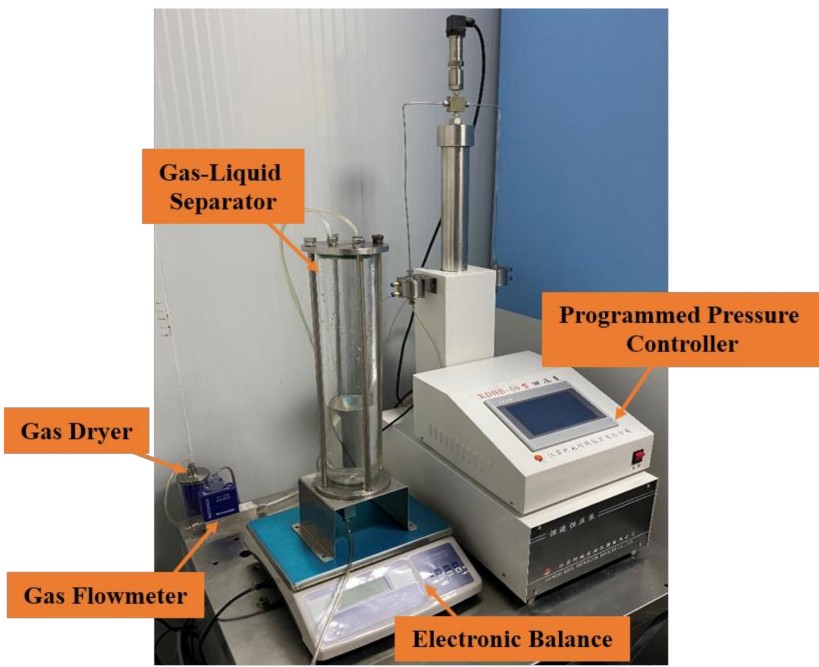

**Figure 7.** Schematic diagram of gas-liquid collection system.

The complete experimental system composed of the above components is shown in Figure 8. Through this device system, the hydrate exploitation in the simulated in situ environment containing hydrate-bearing sediment, overlying strata and overlying seawater can be achieved. The interaction between the strata and hydrate dissociation, and the $CH_4$ leakage during hydrate dissociation can be well reflected. In order to validate the good function of each system of the device, an experimental test was carried out.

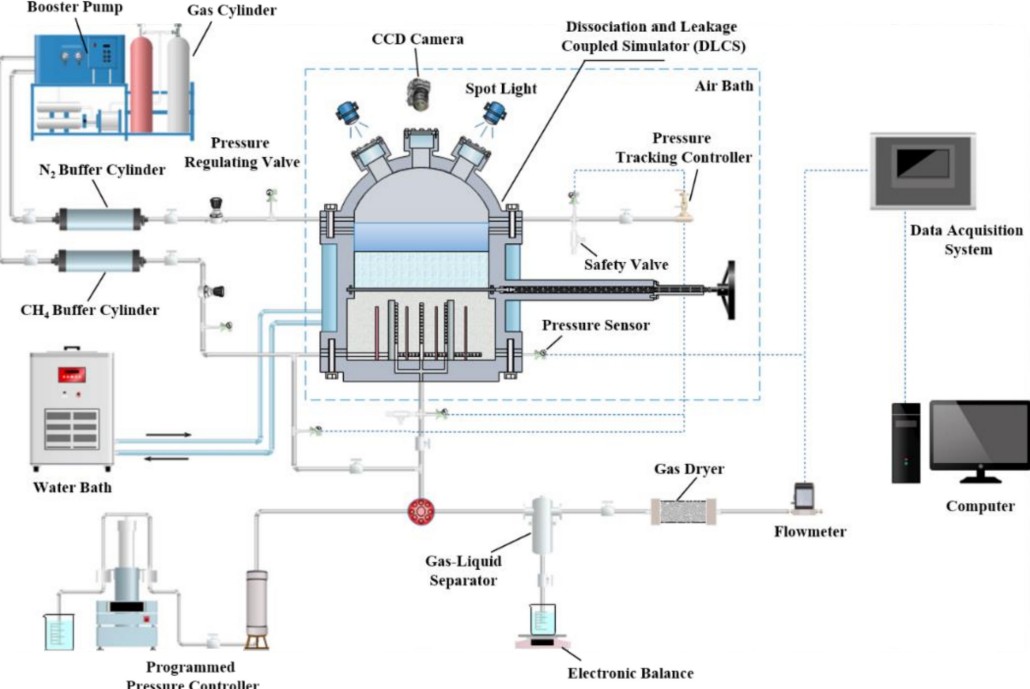

**Figure 8.** Schematic diagram of the complete experimental system.

### 2.2. Experimental Methods

The experiment process mainly includes two parts: hydrate formation and dissociation. The simulated hydrate sediment environment is water saturation condition [37]. The detailed distribution of the measurement points and the configuration of the production wells are shown in Figure 9. In this experiment, only the central well was used for gas production. The DLCS was first cleaned with deionized water, and then the quartz sand with particle size in the range of 330–610 μm was filled in the SC and compacted, in which the detailed particle size distribution is shown in Figure 10. The porosity of the quartz sand sediment was approximately 42.0%. Then, the flashboard was inserted to separate the OC and SC of the reactor. After that, the OC was filled with mixed sediments (including silt, coarse sand, bentonite, kaolin and calcite) and compacted until it was difficult for water to penetrate downward. The mixed sediments were used to simulate the low-permeability overlying strata of hydrate-bearing sediment. Both the OC and SC were then vacuum, and the OC was injected with a certain amount of deionized water to simulate the overlying seawater. After that, $CH_4$ gas with certain pressure was injected into the SC. In the meantime, the OC needs to be pressurized with $N_2$ gas to maintain the pressure balance. Next, deionized water was injected into the SC, while nitrogen into the OC until the pressure in the SC reached 20 MPa. Then, the ambient temperature was adjusted to 8 °C for hydrate formation. $N_2$ in the OC did not form hydrate in this temperature and pressure condition, but only played the role of inert gas to maintain the pressure balance of the system. Every time the pressure of the SC dropped to 15 MPa, deionized water was injected again for the continued hydrate formation. After several times of water injection, the pressure in the SC decreased to 13.2 MPa, and the hydrate formation process finished. The hydrate, gas and water saturation can be calculated according to previous studies [17,38,39].

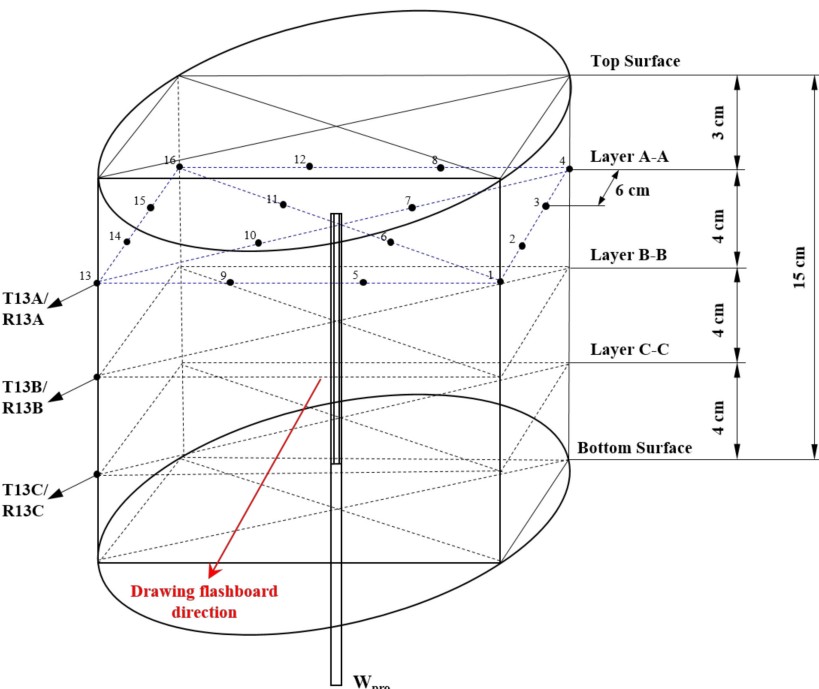

**Figure 9.** Temperature/electric resistance measurement points distribution and well configurations in the DLCS.

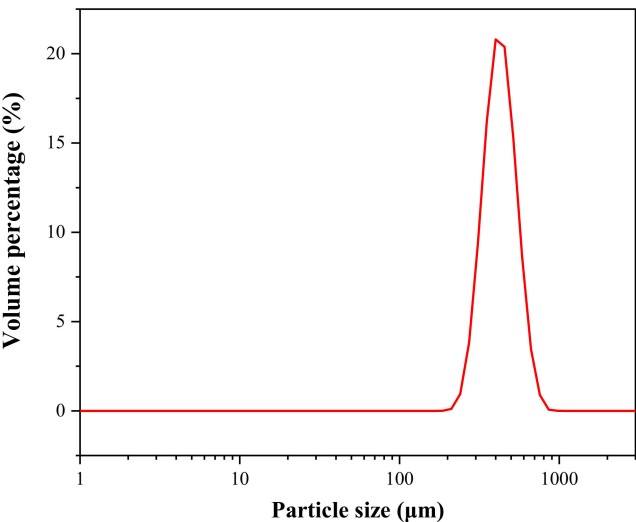

**Figure 10.** Particle size distribution of the quartz sand.

After the complement of the hydrate formation process, the pressures in the OC and SC were adjusted to be the same, and then the flashboard was pulled out to connect the two chambers. In this experiment, the initial hydrate, water and gas saturation in the SC were 0.22, 0.67 and 0.11, respectively. The pressure in the hydrate sediment was reduced from 13.2 MPa to 4.5 MPa at the initial temperature of 8 °C with a pressure reduction rate of 0.5 MPa/min, which was characterized as the depressurizing (DP) stage. After that, the hydrate dissociation continued under constant 4.5 MPa defined as the constant-pressure (CP) stage. When no gas was produced and the system temperature was stable at 8 °C, the experiment ended. The evolutions of temperature and electric resistance can be used to reflect the hydrate dissociation and fluid migration changes in hydrate sediment. The state changes of the overlying strata and overlying seawater were recorded by the CCD. Combined with CCD and gas chromatography, the leakage of $CH_4$ gas from hydrate-bearing sediment to the overlying seawater can be analyzed.

## 3. Experimental Validation for $CH_4$ Hydrate Dissociation

### 3.1. Evolutions of Pressure and Temperatures

Figure 11 presents the system pressure change and the temperature evolutions from the boundary to the center and different horizons of the hydrate-bearing sediment during the depressurization and hydrate dissociation processes. The whole process can be divided into DP stage and CP stage. The dissociation pressure of $CH_4$ hydrate at 8 °C is approximately 5.85 MPa [40], and therefore, a hydrate dissociation stage was included in the DP stage. Due to the Joule–Thomson effect, the temperature may decrease during gas exhaust. However, the temperatures remained almost unchanged before depressurized to the hydrate dissociation pressure in this experiment, even increased slightly. Since there was no other high-temperature heat source in the system, it may be caused by the exothermic reformation of local hydrate caused by fluid disturbance [41]. All the temperatures dropped rapidly after hydrate dissociation, among which the reduced degrees of T4A and T7A were the most obvious. The declined system temperatures were mainly due to the latent heat of phase transition absorbed by dissociated hydrate. The heat mainly derived from the sensible heat of hydrate sediments, overlying strata and overlying seawater. For unevenly distributed hydrate, the temperature will show obvious differences in sediment space due to various amounts of heat adsorption. According to the degree, it can be judged that the hydrate is mainly distributed in the upper hydrate sediments. In the stage of rapid temperature decline, the points T4 in each layer was slightly higher than T7, which showed the movement of the hydrate dissociation front to some extent

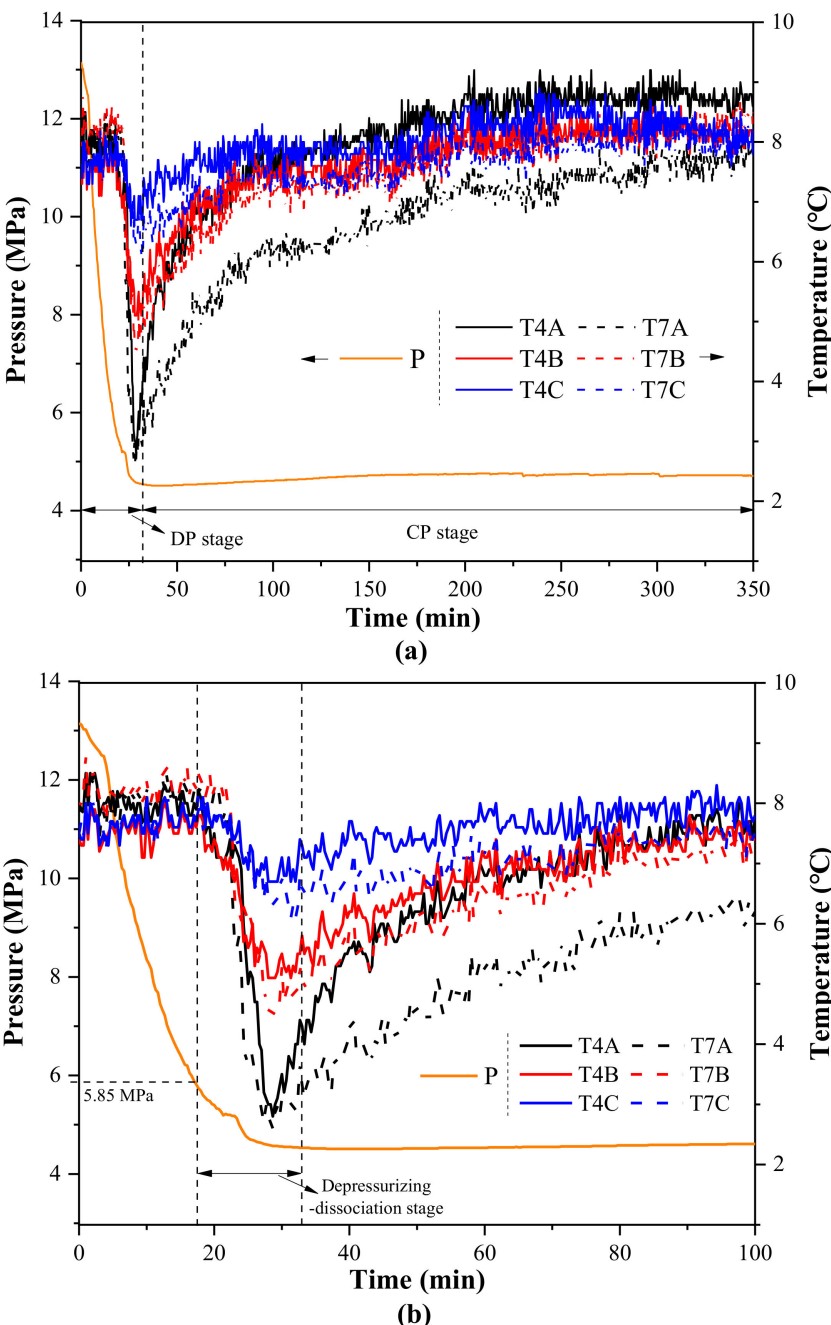

**Figure 11.** System pressure change and temperature evolutions from the boundary to the center and different horizons of the hydrate-bearing sediment during hydrate dissociation. (**a**) Entire hydrate dissociation process. (**b**) The first 100 min for hydrate dissociation.

These temperatures gradually rose after falling to the lowest. This is due to the limited amount of residual hydrate and the low driving force for hydrate dissociation at the same time. Thereby, the heat transfer rate from the environment to the system was greater than the heat absorption rate of hydrate dissociation. In addition, since the ambient heat was transferred from the wall of the reactor to the center, the rise rates of all the points T4 were faster than those of T7 as a whole. The above rules were consistent with many previous reports [42–44]. However, it is worth noting that although the temperature of T4A was distinctly lower than those of T4B and T4C after rapid hydrate dissociation process, its rise rate was the fastest in the subsequent stage, even exceeding the temperatures of other locations after 150 min. This can be mainly attributed to the fact that T4A was closest to the

water bath jacket, in which the heat conduction of water is significantly greater than that of air in the air bath. In addition, the heat conduction from the overlying strata and the probable convective heat transfer from the downward penetration of overlying seawater can also accelerate temperature rise. Therefore, the overlying layer system acted as an additional heat source. This is different from the previous hydrate dissociation studies conducted in closed systems without overlying layers, in which the heat transfer was mainly from the reactor wall to the sediment center [17–20].

The spatial distribution of temperatures in the hydrate-bearing sediment from the starting point to the end point of hydrate dissociation is presented in Figure 12. At 20 min, $CH_4$ hydrate initially dissociated, and the temperature in the whole SC decreased slightly. It can be found that the temperatures around the production well were relatively higher. This is because of the greater turbulence degree, which enhanced gas-liquid contact and hydrate reformation. The overall temperature in the SC dropped to the lowest at about 28 min, and the spatial temperature distribution was almost divided into two areas, of which the low-temperature area was mainly distributed in the said of points T4 and T7 in the upper sediment. One reason is that in the hydrate formation process, hydrate wall climbing phenomenon occurred at the top of points T4 side and induced the migration of a large amount of water, which resulted in seriously inhomogeneous hydrate distribution [45]. In fact, there might also be quantities of $CH_4$ hydrates on the top of the sediment at T13 side. However, during the depressurization and hydrate dissociation stage, the overlying seawater flowed to the production well mainly through this area, and therefore, the gas diffusion restriction caused by high saturation of pore water inhibited the dissociation of $CH_4$ hydrate. In addition, the continuous convective heat transfer supply further slowed down the temperature drop. Following the lowest temperature, all the temperatures in the hydrate sediment gradually rose due to the decreased hydrate dissociation rate and external heat transfer. At 75 min, the overall temperatures of the reservoir were higher than 6 °C, and the low temperature zone extended from the sediment center to the boundary. It should be noted that the temperatures at T13 side were lower than that at 28 min instead, which confirmed that residual $CH_4$ hydrate in this area continued to dissociate during this period. However, due to the low driving force, its dissociation was relatively mild. The heat transferred from the environment could nearly meet the demand for hydrate dissociation, so the temperature decreased slightly. To sum up, the overlying seawater infiltration can significantly influence hydrate dissociation and flow field changes in the reservoir. On the one hand, the downward penetration of seawater brings additional heat supply for hydrate dissociation, but it will also significantly inhibit hydrate dissociation due to diffusion restriction and consequently affect the gas production rate. At 185 min, with the almost complete dissociation of $CH_4$ hydrate, there was only a relatively low temperature zone, the sediment center. At 345 min, the whole reservoir temperature returned to the initial temperature before depressurization. The movement of temperature front was similar to the phenomenon found by Fitzgerald et al. [41] In this early work, they developed a set of classic large-scale hydrate formation/dissociation device with 70 L volume. The hydrate dissociation by central heat stimulation method was studied by using the device. The result showed that the high temperature front moved from the sediment center to the edge during hydrate dissociation process.

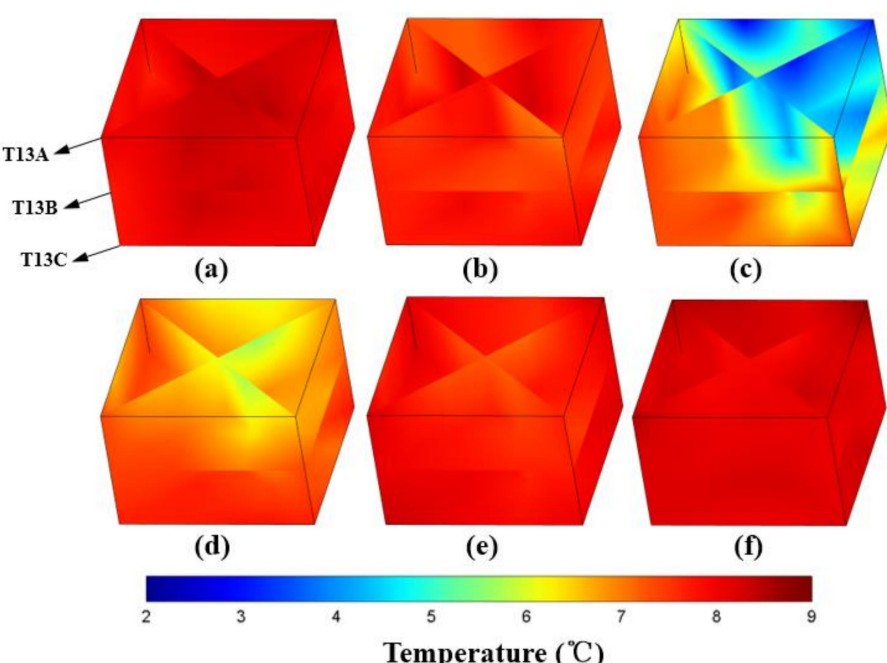

**Figure 12.** Space distributions of temperatures in the hydrate-bearing sediment from the starting point to the end point of hydrate dissociation. (**a**) 0 min, (**b**) 20 min, (**c**) 28 min, (**d**) 75 min, (**e**) 185 min, (**f**) 345 min.

### 3.2. Electric Resistance Evolution Characteristic

The evolution of $R_t/R_0$ ratios for R4 and R7 in the three layers during the whole gas production process is shown in Figure 13. The $R_0$ represents the initial resistance value before depressurization, and $R_t$ represents the resistance value at time t. The dotted lines A and B in the graph represent the time for initial hydrate dissociation and the end of depressurization process, respectively. The electric resistance decreased as a whole in the initial depressurization period. This is because fluid migration changed the gas/water/hydrate contents between the pair of electrodes of a single electric resistance. We speculate that free water replaced suspended hydrates. During the period between time A and B, the resistance values of R7A and R4B rose significantly, and the R4A and R7B also increased slightly. This indicated a large amount of free gas generated from dissociated $CH_4$ hydrate, which displaced part of the free water between the electrodes and resulted in the sharp increase in electric resistance. In the early stage of CP stage, each resistance value showed nearly continuous change trend. The sudden increase and decrease of resistance values during this period can be mainly attributed to the filling of gas and water between the pair of electrodes, respectively. Therefore, the movement of flow field at each position can be deduced through the change of resistance value. It should be mentioned that due to the high recording frequency (10 s for once) of resistance data in this experimental test, some resistance values were not collected after 150 min. Consequently, the stepped trend phenomenon occurred in Figure 13. In future experiments, the resistance acquisition time will be adjusted. After the complete hydrate dissociation, each resistance value no longer obviously changed.

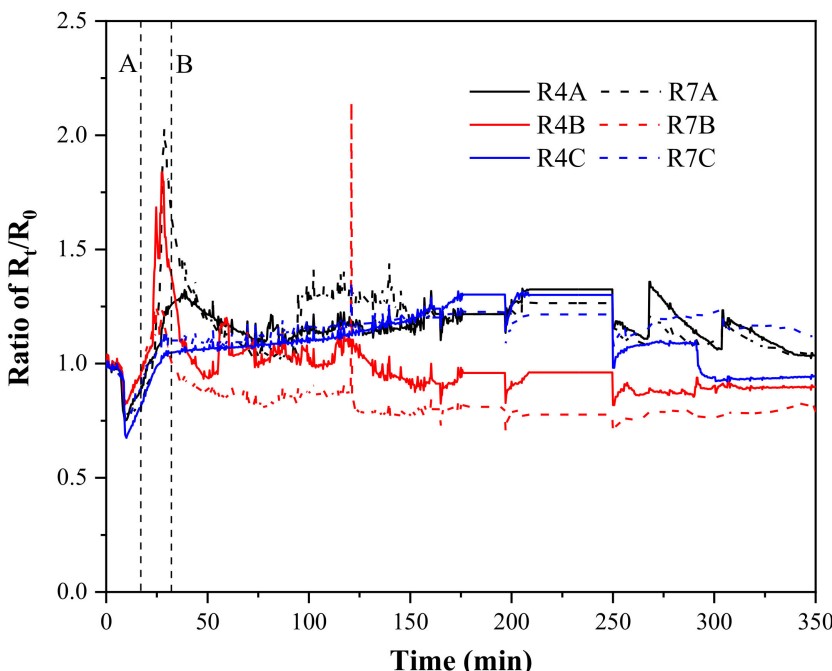

**Figure 13.** Evolution of $R_t/R_0$ ratios for R4 and R7 from the boundary to the center and different horizons of the hydrate-bearing sediment during the whole gas production process.

The space distributions of $R_t/R_0$ ratio in the hydrate-bearing sediment from the starting point to the end point of hydrate dissociation is displayed in Figure 14. Except for R10 with obvious decrease, the other resistance values increased or decreased slightly during the initial hydrate dissociation process. The decline of R10 may be because overlying seawater flowed to production well mainly path through this area. The increase of water content between each pair of electrodes of points R10 resulted in the reduction of the resistance values. This phenomenon is consistent with our above inference in Figure 12. The resistance values in the R4 and R7 area rose at the stage of rapid hydrate dissociation around 28 min. This may be related to the high hydrate saturation which produced a large amount of tiny $CH_4$ gas bubbles during the dissociation process. After the hydrate dissociation rate slowed down and the sediment temperature gradually rose with the heat transfer from ambient temperature, the resistance value in the sediment center showed a decreasing trend. We consider that this is because the residual water was no longer massively expelled or entrained out of the system with the decrease of gas production rate, and abundant free water deposited or adhered around this area. After the spatial temperatures in the whole sediment almost returned to 8 °C at 345 min, the resistances did not return to the original values, but decreased as a whole. The average resistance at the center of the sediment was about half of the original value relative to that in 0 min, which implied that most of the free water in the sediment is mainly concentrated in the sediment center. In conclusion, the combination of temperature and resistance in spatial distribution can reflect the changes of hydrate dissociation, fluid migration path and fluid distribution in porous sediments.

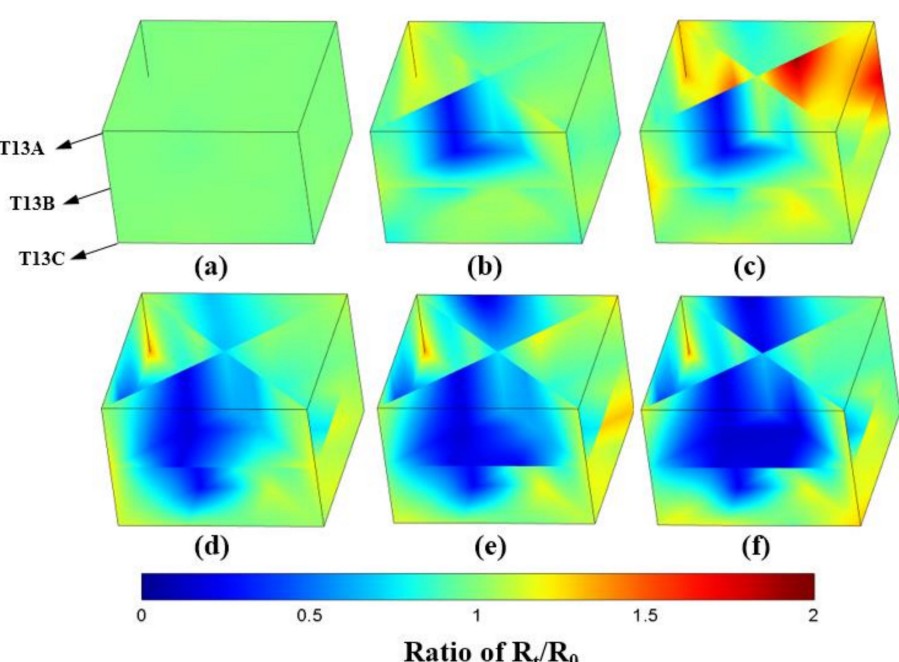

**Figure 14.** Space distributions of $R_t/R_0$ ratios in the hydrate-bearing sediment from the starting point to the end point of hydrate dissociation. (**a**) 0 min, (**b**) 20 min, (**c**) 28 min, (**d**) 75 min, (**e**) 185 min, (**f**) 345 min.

*3.3. Production Behaviors*

Figure 15 presents the changes in gas and water production volumes during the whole depressurization and hydrate dissociation process. It can be observed that before the pressure dropped to the hydrate dissociation pressure, the product was mainly water with only a small amount of gas being discharged. This is because the system was water saturated and the gas content was only 11%. The gas produced in this stage was mainly free gas with a very small amount of dissolved gas. After the pressure dropped below 5.85 MPa, the hydrate began to dissociate and led to the decrease of reservoir temperature at 20 min, as shown in Figure 11. However, the gas production rate did not obviously increase, and the gas produced by hydrate dissociation was temporarily bound inside the sediment in the period. At about 20 min, the gas production rate was significantly accelerated, and the water production rate gradually slowed down. During the period of 20–33 min, the produced gas was mainly derived from $CH_4$ hydrate dissociation. The large amount of gas produced in this stage corresponded to the rapid decline of temperature in Figure 11. There was almost no water production between 50 min and 175 min in the CP stage. During this period, gas from dissociated $CH_4$ hydrate was slowly produced. The changed gas production rate was similar to the research in a closed system conducted by Fitzgerald et al. [41] In their experiment, the initial $CH_4$ production yield caused by hydrate dissociation first accelerated and then gradually slowed down. However, the decrease in gas production rate was mainly due to the reduced hydrate saturation in their experiment. In our experiment, in addition to the small quantity of residual hydrate, temperature was also an important factor limiting hydrate dissociation. The sudden step rise of gas production occurred at 225 min may be due to the temporary blockage of the production well. After 350 min, the hydrate was completely dissociated, and the final gas and water production were 96 L and 10.7 L, respectively. Since the total amount of water injected into the SC was only 4 L, the extra water came from the overlying seawater. This demonstrates the inference of overlying seawater infiltration analyzed above.

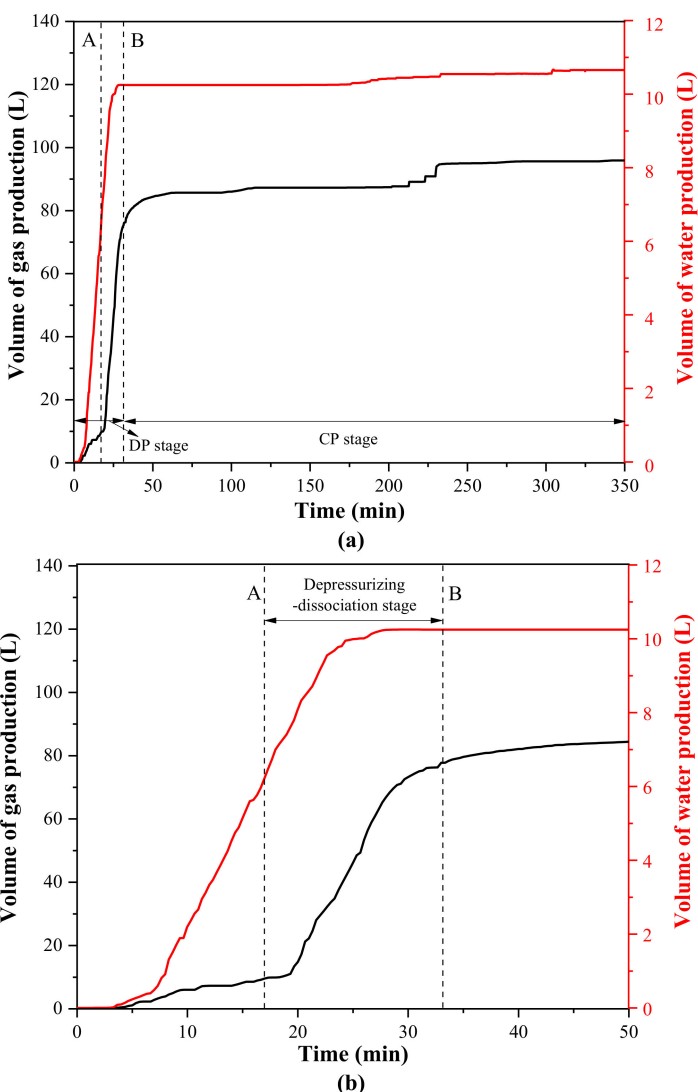

**Figure 15.** Changes of volumes of gas and water production during depressurization and hydrate dissociation processes. (**a**) Entire hydrate dissociation process. (**b**) The first 50 min for hydrate dissociation.

### 3.4. Morphological Change of Overlying Strata before and after Hydrate Dissociation

The morphologies of the overlying strata before and after hydrate dissociation are shown in Figure 16. The rod in the middle of the picture is a temperature sensor located at the overlying seawater layer. The overlying strata was complete without obvious cracking before depressurization. However, it can be seen that an obvious crack appeared on the strata after hydrate dissociation, indicating local collapse of the overlying strata. It should be mentioned that the orientation of the graph corresponds to Figure 9, that is, the fracture was nearly above points T13/R13 and T14/R14. In the depressurization process, the overlying seawater tended to flow down to the vicinity of points T13/R13 of the hydrate reservoir through the fault, which is in line with the phenomenon in Figure 12. Although the obvious crack phenomenon occurred, there was no $CH_4$ leakage up to the overlying seawater according to leakage bubble observation and gas content analysis. It may be related to the massive accumulation of $CH_4$ hydrate near points T4/R4. On the other hand, it may be attributed to the constant rate depressurization method used in this experiment. Nevertheless, the collapse of the overlying strata should be avoided. Otherwise, the downward flow of seawater into hydrate reservoir will not only reduce the gas/water ratio, but also inhibit the hydrate dissociation.

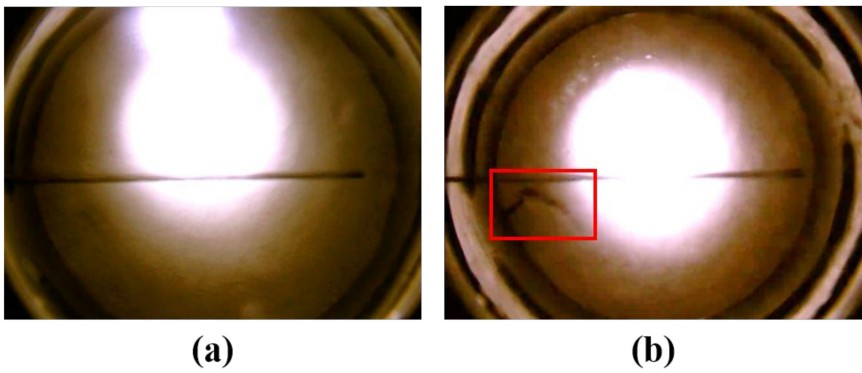

**Figure 16.** Morphological change of the overlying strata. (**a**) Before depressurization. (**b**) After hydrate dissociation.

## 4. Conclusions

This work describes a coupled simulation device for the study of methane hydrate dissociation and methane leakage. The functions of the device were verified by an experimental test. The conclusions obtained in this study mainly include the following:

1.  In the new device, the in situ natural environment containing hydrate reservoir, over lying strata and overlying seawater could be well simulated by utilizing the elaborately designed mobile separation flashboard. The evolutions of pressure field, temperature field and electric field in the hydrate dissociation process can be well reflected through the integrated temperature, electric resistance and pressure measuring points in the hydrate reservoir systems.
2.  The real-time recording of overlying strata morphology changes and $CH_4$ bubble leakage can be realized through the overburden monitoring system. In addition, the $CH_4$ concentration in overlying seawater and gas phase space can be measured to further judge whether there is methane leakage.
3.  The experimental test showed that all functions of the system can be operated normally. According to the experimental result, the downward-flow overlying seawater significantly affected the hydrate dissociation, multi-field evolution and gas production characteristics. The state change of the overlying strata before and after depressurization indicated that there was local collapse of the overlying strata during the hydrate dissociation process.
4.  Despite the obvious collapse of the overlying strata, $CH_4$ leakage phenomenon did not occur in this test, which may be related to the hydrate distribution or the selection of hydrate exploitation methods in this test. However, it is necessary to avoid the occurrence of overburden fracture channel and seawater downward flow, otherwise it will significantly affect gas production and energy efficiency.

**Author Contributions:** Conceptualization, J.F. and Y.W. (Yi Wang); methodology, J.F., Y.X. and L.S.; software, W.H.; validation, L.S., J.W. and W.H.; formal analysis, J.F.; investigation, B.P.; resources, Y.W. (Yujun Wang); data curation, J.W.; writing—original draft preparation, Y.X.; writing—review and editing, J.F.; visualization, Y.W. (Yi Wang); supervision, J.F.; project administration, J.F.; funding acquisition, J.F. All authors have read and agreed to the published version of the manuscript.

**Funding:** Financial support received from the National Natural Science Foundation of China (42022046, 52122602), the National Key Research and Development Program of China (2021YFF0502300), the Key Special Project for Introduced Talents Team of Southern Marine Science and Engineering Guangdong Laboratory (Guangzhou) (GML2019ZD0403 and GML2019ZD0401), Guangdong Natural Resources Foundation (GDNRC [2022]45), Guangzhou Science and Technology Project (202102020971), the Natural Science Foundation of Guangdong Province (2022A1515010590) and China Postdoctoral Science Foundation (2021M690726) are gratefully acknowledged.

**Institutional Review Board Statement:** Not applicable.

**Informed Consent Statement:** Not applicable.

**Data Availability Statement:** Data associated with this research are available and can be obtained by contacting the corresponding author.

**Acknowledgments:** Financial support received from the National Natural Science Foundation of China (42022046, 52122602), the National Key Research and Development Program of China (2021YFF0502300), the Key Special Project for Introduced Talents Team of Southern Marine Science and Engineering Guangdong Laboratory (Guangzhou) (GML2019ZD0403 and GML2019ZD0401), Guangdong Natural Resources Foundation (GDNRC [2022]45), Guangzhou Science and Technology Project (202102020971), the Natural Science Foundation of Guangdong Province (2022A1515010590) and China Postdoctoral Science Foundation (2021M690726) are gratefully acknowledged.

**Conflicts of Interest:** The authors declare no conflict of interest.

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
