# Peer review of "Coupled Simulation of Hydrate-Bearing and Overburden Sedimentary Layers to Study Hydrate Dissociation and Methane Leakage"

_jmse, doi:10.3390/jmse10050668_

Round 1
Reviewer 1 Report
If compared with the previous submission, the article appears strongly improved. The introduction contains more precise and detailed information; the experimental apparatus is better described and the methodology followed too. Also the additions provided in the experimental section allow to better define the reliability of results and favours the comparison with the current research available in literature.
For that reason, I recommend the publication of this article, as it is.
Author Response
Response to the editor and reviewers
We are very grateful to the editor and reviewers for giving us opportunities to revise our manuscript. We appreciate for their positive and constructive comments and suggestions on our manuscript entitled “Coupled simulation device of hydrate-bearing and overburden layers to study hydrate dissociation and methane leakage”.
Editor comments:  
Comment: Dear Dr. Feng: Thank you again for you submission, I am very happy to report that I have recommended your contribution for publication after you deal with a few minor editorial concerns as provided by Reviewer-2. I would recommend changing the use of the term "law" to maybe "controls," which would be more appropriate. Congratulations! Tim Collett
Reply: Thanks for the editor’s valuable suggestion and recommendation very much. We have improved our manuscript according to the editor and reviewers’ suggestions. In the revised manuscript, the “occurrence law” has been changed into “influential factors”. In addition, the title has been changed, the writing has been improved, and some descriptions have been modified.
Comments and Suggestions for Authors:
Reviewer #1:
Comment: If compared with the previous submission, the article appears strongly improved. The introduction contains more precise and detailed information; the experimental apparatus is better described and the methodology followed too. Also the additions provided in the experimental section allow to better define the reliability of results and favours the comparison with the current research available in literature.
For that reason, I recommend the publication of this article, as it is.
Reply: Thanks for the reviewer’s kind comments and constructive suggestions in the first review very much. Those proposed suggestions are really very helpful for the improvement of our manuscript.
Reviewer #2:
Comment: This is a second review of a manuscript regarding a coupled simulator for hydrate-bearing sediments and process and impacts of dissociation and potential methane leakage. The authors appear to have put considerable effort to address the reviewers’ comments. I think it can be published with the consideration below address.
Reply: Thanks for the reviewer’s kind comments and constructive suggestions very much. We have improved our manuscript according to reviewer’s suggestions. In the revised manuscript, the title has been changed, the writing has been improved, and some describes have been modified.
Suggestion: To avoid confusion, I would change the word simulator and use of the word modeled. I can see very easy that someone who starts to read this is expecting a simulation model to accompany the experimental work. Not sure the best way, maybe “Coupled measurements…”.
Reply: Thanks for the reviewer’s valuable advice. In the hydrate industry, "simulator" may generally refer to experimental reactor, but we did not realize that in many other industries, the word also refers to simulation model. On the other hand, since "model" usually refers to mathematical model in hydrate industry, and the words "coupled measurements" might focus more on measurement methods than new equipment, we change the word "simulator" to "simulation device" in the revised title after deep consideration. We are very grateful to the reviewer’s valuable suggestions.
Suggestion: As I mentioned before, be careful for some claims of NGH (Line 41-42). NGH are mainly methane and (even if biogenic) is extracted from the subsurface and contributes to the GHG emissions just like natural gas. Relative to coal and crude oil, emissions are low but if is still a significant source of emissions to the atmosphere compared to renewables, etc.
Reply: Thanks for the reviewer’s reminder and valuable advice. We have revised the improper claim. “NGH is regarded as an important follow-up energy in the 21st century” in the original manuscript has been changed in to “NGH has attracted the attention of a large number of scholars”. (Line 41-42 in the revised manuscript)
Suggestion: The experimental apparatus is interesting and useful to see the extra detail.
Reply: Thanks for the reviewer’s recognition. We have supplemented the main size parameters of the device. “The effective volumes of the OC and SC are 19.9 L and 10.3 L respectively, in which the volume of the upper cap is 8.5 L. The maximum sediment filling height of the SC is 15 cm, the height of the intermediate reactor body contained in the OC is 20 cm, and the filling height of the overlying layer in the OC in this study was 10 cm.” (Line 127-130 in the revised manuscript). In future work, more interesting experimental phenomena and useful conclusions will be obtained by using the device.
Suggestion: Line104-116: What are the criteria for scale-up? Medium or large size compared to what? It would be good to address the boundary effect and what determines the required sample size to give a representative result to nature.
Reply: Thanks for the reviewer’s professional suggestion. The scale-up mentioned in this article mainly refers to the enlargement of the effective volume of the reactor. On the one hand, it is to relatively reduce the unavoidable boundary effect. On the other hand, the increased volume will make the distribution and state of accumulated hydrate, and multi-field coupling in the hydrate dissociation process more complex. The hydrate sediment could therefore be relatively closer to the condition of natural sediment. However, the reported reactor scale at present may still not give an answer as to what size of sample can give representative results to nature. Because the experiments reported so far have not been completely or almost close to the natural hydrate sediment condition. A larger reactor may be needed to support the conclusion.
The medium and large scale mentioned in this article are mainly defined according to the volume, which is compared with the reported NGH reservoir formation/dissociation simulation device, but in fact, there are no criteria for judging the scale. As far as we know, the simulators currently used for hydrate research include (1) small simulators with a volume less than 1 L, usually combined with CT, NMR, Raman and other precision detection methods; (2) Small and medium-sized simulators with volume less than 10 L, which are mainly equipped with temperature and pressure detection means; (3) Pilot-scale simulator with the volume greater than 100 L; (4) A giant simulator with a volume of 1710 L; (5) For simulators between 10-100 L, we classify them as medium-sized. However, since there is no standard division, the classification has subjective consciousness to a certain extent. Therefore, we have revised the description as follows:
“Due to the great dimension of real NGH deposits, hydrate simulator is gradually developed to large scale through increasing the effective volume. According to reports, the largest reactor has been developed to 1710 L. With the enlarged scale, relatively reduced boundary effect, and more complex hydrate formation/dissociation state, the experimental condition could be regarded as closer to the in-situ hydrate-bearing sediment environment [10, 33, 36]. For the newly built simulator in this study, considering the increasing difficulty of design and experimental operation with the enlargement of the size, the effective volume of 30.2 L was adopted.” (Line 114-121 in the revised manuscript)
Other comments/questions:
The English has improved. There are still some issues but much better than the previous version.
In the abstract, What does “occurrence law” mean?
Line 34: Maybe “water molecules which form cages and trap gas molecules”. If not it seems the water and gas molecules are h-bonded together.
Reply: Thanks for the reviewer’s careful work and valuable suggestion. The “occurrence law” in the abstract in the original manuscript was originally intended to express what the occurrence process is like and what factors will affect the CH4 leakage characteristics. We have modified the “occurrence law” into “influential factors” in the revised abstract.
In addition, “Natural gas hydrate (NGH), also known as combustible ice, is a non-stoichiometric crystalline compound composed of hydrogen-bonded water molecules and gas molecules” in the original manuscript has been change into “Natural gas hydrate (NGH), also known as combustible ice, is a non-stoichiometric crystalline compound, in which water molecules form cages and trap gas molecules”. (Line 34 in the revised manuscript).

Reviewer 2 Report
This is a second review of a manuscript regarding a coupled simulator for hydrate-bearing sediments and process and impacts of dissociation and potential methane leakage. The authors appear to have put considerable effort to address the reviewers’ comments. I think it can be published with the consideration below address.
To avoid confusion, I would change the word simulator and use of the word modeled. I can see very easy that someone who starts to read this is expecting a simulation model to accompany the experimental work. Not sure the best way, maybe “Coupled measurements…”.
As I mentioned before, be careful for some claims of NGH (Line 41-42). NGH are mainly methane and (even if biogenic) is extracted from the subsurface and contributes to the GHG emissions just like natural gas. Relative to coal and crude oil, emissions are low but if is still a significant source of emissions to the atmosphere compared to renewables, etc.
The experimental apparatus is interesting and useful to see the extra detail.
Line104-116: What are the criteria for scale-up? Medium or large size compared to what? It would be good to address the boundary effect and what determines the required sample size to give a representative result to nature.
Other comments/questions:
- The English has improved. There are still some issues but much better than the previous version.
- In the abstract,
- What does “occurrence law” mean?
- Line 34: Maybe “water molecules which form cages and trap gas molecules”. If not it seems the water and gas molecules are h-bonded together.
Author Response
Response to the editor and reviewers
We are very grateful to the editor and reviewers for giving us opportunities to revise our manuscript. We appreciate for their positive and constructive comments and suggestions on our manuscript entitled “Coupled simulation device of hydrate-bearing and overburden layers to study hydrate dissociation and methane leakage”.
Editor comments:  
Comment: Dear Dr. Feng: Thank you again for you submission, I am very happy to report that I have recommended your contribution for publication after you deal with a few minor editorial concerns as provided by Reviewer-2. I would recommend changing the use of the term "law" to maybe "controls," which would be more appropriate. Congratulations! Tim Collett
Reply: Thanks for the editor’s valuable suggestion and recommendation very much. We have improved our manuscript according to the editor and reviewers’ suggestions. In the revised manuscript, the “occurrence law” has been changed into “influential factors”. In addition, the title has been changed, the writing has been improved, and some descriptions have been modified.
Comments and Suggestions for Authors:
Reviewer #1:
Comment: If compared with the previous submission, the article appears strongly improved. The introduction contains more precise and detailed information; the experimental apparatus is better described and the methodology followed too. Also the additions provided in the experimental section allow to better define the reliability of results and favours the comparison with the current research available in literature.
For that reason, I recommend the publication of this article, as it is.
Reply: Thanks for the reviewer’s kind comments and constructive suggestions in the first review very much. Those proposed suggestions are really very helpful for the improvement of our manuscript.
Reviewer #2:
Comment: This is a second review of a manuscript regarding a coupled simulator for hydrate-bearing sediments and process and impacts of dissociation and potential methane leakage. The authors appear to have put considerable effort to address the reviewers’ comments. I think it can be published with the consideration below address.
Reply: Thanks for the reviewer’s kind comments and constructive suggestions very much. We have improved our manuscript according to reviewer’s suggestions. In the revised manuscript, the title has been changed, the writing has been improved, and some describes have been modified.
Suggestion: To avoid confusion, I would change the word simulator and use of the word modeled. I can see very easy that someone who starts to read this is expecting a simulation model to accompany the experimental work. Not sure the best way, maybe “Coupled measurements…”.
Reply: Thanks for the reviewer’s valuable advice. In the hydrate industry, "simulator" may generally refer to experimental reactor, but we did not realize that in many other industries, the word also refers to simulation model. On the other hand, since "model" usually refers to mathematical model in hydrate industry, and the words "coupled measurements" might focus more on measurement methods than new equipment, we change the word "simulator" to "simulation device" in the revised title after deep consideration. We are very grateful to the reviewer’s valuable suggestions.
Suggestion: As I mentioned before, be careful for some claims of NGH (Line 41-42). NGH are mainly methane and (even if biogenic) is extracted from the subsurface and contributes to the GHG emissions just like natural gas. Relative to coal and crude oil, emissions are low but if is still a significant source of emissions to the atmosphere compared to renewables, etc.
Reply: Thanks for the reviewer’s reminder and valuable advice. We have revised the improper claim. “NGH is regarded as an important follow-up energy in the 21st century” in the original manuscript has been changed in to “NGH has attracted the attention of a large number of scholars”. (Line 41-42 in the revised manuscript)
Suggestion: The experimental apparatus is interesting and useful to see the extra detail.
Reply: Thanks for the reviewer’s recognition. We have supplemented the main size parameters of the device. “The effective volumes of the OC and SC are 19.9 L and 10.3 L respectively, in which the volume of the upper cap is 8.5 L. The maximum sediment filling height of the SC is 15 cm, the height of the intermediate reactor body contained in the OC is 20 cm, and the filling height of the overlying layer in the OC in this study was 10 cm.” (Line 127-130 in the revised manuscript). In future work, more interesting experimental phenomena and useful conclusions will be obtained by using the device.
Suggestion: Line104-116: What are the criteria for scale-up? Medium or large size compared to what? It would be good to address the boundary effect and what determines the required sample size to give a representative result to nature.
Reply: Thanks for the reviewer’s professional suggestion. The scale-up mentioned in this article mainly refers to the enlargement of the effective volume of the reactor. On the one hand, it is to relatively reduce the unavoidable boundary effect. On the other hand, the increased volume will make the distribution and state of accumulated hydrate, and multi-field coupling in the hydrate dissociation process more complex. The hydrate sediment could therefore be relatively closer to the condition of natural sediment. However, the reported reactor scale at present may still not give an answer as to what size of sample can give representative results to nature. Because the experiments reported so far have not been completely or almost close to the natural hydrate sediment condition. A larger reactor may be needed to support the conclusion.
The medium and large scale mentioned in this article are mainly defined according to the volume, which is compared with the reported NGH reservoir formation/dissociation simulation device, but in fact, there are no criteria for judging the scale. As far as we know, the simulators currently used for hydrate research include (1) small simulators with a volume less than 1 L, usually combined with CT, NMR, Raman and other precision detection methods; (2) Small and medium-sized simulators with volume less than 10 L, which are mainly equipped with temperature and pressure detection means; (3) Pilot-scale simulator with the volume greater than 100 L; (4) A giant simulator with a volume of 1710 L; (5) For simulators between 10-100 L, we classify them as medium-sized. However, since there is no standard division, the classification has subjective consciousness to a certain extent. Therefore, we have revised the description as follows:
“Due to the great dimension of real NGH deposits, hydrate simulator is gradually developed to large scale through increasing the effective volume. According to reports, the largest reactor has been developed to 1710 L. With the enlarged scale, relatively reduced boundary effect, and more complex hydrate formation/dissociation state, the experimental condition could be regarded as closer to the in-situ hydrate-bearing sediment environment [10, 33, 36]. For the newly built simulator in this study, considering the increasing difficulty of design and experimental operation with the enlargement of the size, the effective volume of 30.2 L was adopted.” (Line 114-121 in the revised manuscript)
Other comments/questions:
The English has improved. There are still some issues but much better than the previous version.
In the abstract, What does “occurrence law” mean?
Line 34: Maybe “water molecules which form cages and trap gas molecules”. If not it seems the water and gas molecules are h-bonded together.
Reply: Thanks for the reviewer’s careful work and valuable suggestion. The “occurrence law” in the abstract in the original manuscript was originally intended to express what the occurrence process is like and what factors will affect the CH4 leakage characteristics. We have modified the “occurrence law” into “influential factors” in the revised abstract.
In addition, “Natural gas hydrate (NGH), also known as combustible ice, is a non-stoichiometric crystalline compound composed of hydrogen-bonded water molecules and gas molecules” in the original manuscript has been change into “Natural gas hydrate (NGH), also known as combustible ice, is a non-stoichiometric crystalline compound, in which water molecules form cages and trap gas molecules”. (Line 34 in the revised manuscript).

This manuscript is a resubmission of an earlier submission. The following is a list of the peer review reports and author responses from that submission.
Round 1
Reviewer 1 Report
The manuscript presents the design of a device to simulate the dissociationprocess of methane hydrates. In this case it is mentioned that the evolution
of temperature, pressure, electric field, morphological changes of the crystal
and methane concentration in the effluents during the process can be
monitored. However, in the results not all these variables are shown, and the
discussion focuses the analysis on temperature and pressure. It is not clear if
saline water conditions are used in the experimentation that reflect the real
conditions where the design is intended to be applied. Therefore, further discussion of these results is necessary.
Reviewer 2 Report
This paper presents methane leakage as a key bottleneck for exploitation of methane in gas hydrates in use for commercial development. While an issue, I cannot agree that this issue is fundamental in the commercial side of the NGH exploitation. There is scientific merit but as written it presents a paper that will not be utilized. There needs to be a focus without overselling the product. I would suggest that the paper be restructured to focus on the lab work done.
As a general comment, please have someone review the English. It is okay but a quick check with someone could improve the paper a lot for people who find it and start to read it.
NGH is interesting but not the “most potential alternative energy of the 21st century”
I would be happy to review an updated manuscript. In the present form, it goes too far in my opinion and needs major revision.
Reviewer 3 Report
After a careful revision of the manuscript, I decided to propose major revision.
These are my comments:
- There are numerous typos and grammar errors along the whole text, from the introduction to the conclusions.
- The first part of the introduction must be re-written. The grammar is not adequate and some information should be controlled. For instance, the percentage of marine hydrate reservoirs is equal to 97% of the total and not to 90% (see for instance: J. Pet. Sci. Eng. 205 (2021) 108895). I suggest to replace this information or, better, to insert both of them in the text.
- Also the estimation of 3x10^15 m3 is not certain. Several estimeations have been produced, which may differ also about 2-3 orders of magnitude from each other. Also here, some specification more would be appropriate.
- The experimental apparatus is highly interesting in my opinion. However: i) why a so larger volume was chosen?; ii) which kind of pressure and temperature sensors were used?; iii) what about their accuracy?; iv) also the main properties of the air bath and the circulator bath should be mentioned.
- The size of sand is 330 - 610. Can you specify the size distribution?
- What about the porosity of sand?
- The methodology should be inserted in Section 2 instead of Section 3.
- In Section 3.2.1, a diagram describing the pressure-temperature trend during hydrate dissociation should be inserted and discussed.
- Some subtitle is not in the correct format; revise it.
- The stepped trend of temperatures, which appeared from minute 150 in Figure 13, should be carefully discussed in the text.
- Figure 15 is not meaningful in this form, because there are no evaluable differences between picture A and picture B.
- Finally, the article mainly deals with the description of a new lab-scale experimental apparatus and does not provide relevant scientific contributions. In this form, the scientific content is not enough in my opinion. I suggest to the authors to compared the first results they obtained with this reactor and discussed in the text, with similar experiments carried out with different experimental apparatuses, having similar or different sizes (see (J. Pet. Sci. Eng. 94-95 (2012) 19-27). A similar discussion would be useful for other researchers and would increase the suitability of this article for publication.